# The Evolution of Artificial Intelligence in the Digital Economy: An Application of the Potential Dirichlet Allocation Model

**Chunyi Shan [1], Jun Wang [1,2,\*] and Yongming Zhu [1,3]**

1   School of Management, Zhengzhou University, Zhengzhou 450001, China
2   Institute of Big Data Science, Zhengzhou University of Aeronautics, Zhengzhou 450046, China
3   Yellow River Institute for Ecological Protection & Regional Coordinated Development, Zhengzhou University, Zhengzhou 450001, China
\*   Correspondence: zzwjun@126.com

**Abstract:** The most critical driver of the digital economy comes from breakthroughs in cutting-edge technologies such as artificial intelligence. In order to promote technological innovation and layout in the field of artificial intelligence, this paper analyzes the patent text of artificial intelligence technology using the LDA topic model from the perspective of the patent technology subject based on Derwent patent data. The results reveal that AI technology is upgraded from chips, sensing, and algorithms to innovative platforms and intelligent applications. Proposed countermeasures are necessary to advance the digitalization of the global economy and to achieve economic globalization in terms of industrial integration, building ecological systems, and strengthening independent innovation.

**Keywords:** artificial intelligence; digital economy; technology evolution; LDA model

## 1. Introduction

Artificial intelligence (AI) is a new technical science that studies and develops theories, methods, technologies, and application systems for simulating, extending, and expanding human intelligence [1]. In recent years, various technological powers worldwide have introduced relevant strategies to promote the development of artificial intelligence. The EU's "The Digital Europe Programme", Japan's "Artificial Intelligence/Big Data/Internet of Things/Network Security Integrated Project", and the US National Artificial Intelligence Research and Development Strategic Plan have incorporated artificial intelligence into national strategies. In May 2016, China National Development & Reform Commission and Ministry of Science & Technology issued the "Three-Year Action Plan for Internet & Artificial Intelligence". This plan details efforts to develop several artificial intelligence key core technologies, strengthen artificial intelligence application innovation, promote artificial intelligence in key areas of the national economy and society, and other development plans. The State Council (2017) issued the "New Generation Artificial Intelligence Development Plan," which clarified the development strategy goals and six key tasks for developing a new generation of artificial intelligence. It opened a new journey of artificial intelligence innovation development in China. In 2020, the Alliance released a white paper on AI, emphasizing increased funds in research and the application of AI technology.

Artificial intelligence has been widely used in agriculture [2,3], geotechnical engineering [4,5], environmental science [6], and other fields. Therefore, it is important to explore the evolutionary process of AI in the era of the digital economy to indicate the next stage of AI development. This paper analyzes the patent text of artificial intelligence technology through the LDA topic model from the perspective of the patent technology subject, based on Derwent patent data. It is essential for China to carry out technological layout, discover the shortcomings of artificial intelligence technology, and promote technological innovation in artificial-intelligence-related fields.

## 2. Research Background

The patent literature is the carrier of knowledge and technology, containing 90–95% of the world's science and technology information. It has become the best source of information for studying technological innovation and predicting future technology development trends [7,8]. Due to the easy access to patent documents and its aptness for qualitative and quantitative research, various researchers have explored patent technology from a large number of patent documents for their supporting data [9]. The technology evolution analysis is an important research method for patent text analysis, which plays an important role in supporting national science, technology planning, and decision making in enterprise development [10,11]. The LDA (latent Dirichlet allocation) topic model is a three-layer Bayesian calculation used to obtain a mixed text. Its theme and documents comprise a vocabulary with different probability distributions [12]. The LDA topic model is widely used for text topic mining. For semi-structured documents, such as patent documents, the LDA topic model can be used for topic mining to identify the subject words implied in the text, thus helping analyze the evolution of the technology [13].

In recent years, many scholars have conducted research on artificial intelligence. Marvin Minsky [14] defined and conducted theoretical research on this topic. Pollack [15] proposed a modern approach to the "Unified Theory of Artificial Intelligence". LeCun et al. [16] argued that deep learning powers many aspects of modern society, which triggered the explosive growth of artificial intelligence. Guzman et al. [17] reported that machine learning uses data and algorithms to analyze large amounts of data to make decisions and predict behavior. Some artificial intelligence experts, such as Guo et al. [18], argue that current machine learning systems operate almost exclusively in statistical models or model-free modes, preventing breakthroughs in effective artificial intelligence techniques. Pareek et al. [19] argue that huge breakthroughs have been made in artificial intelligence regarding video recognition, speech recognition, and machine translation through deep learning. Giczy et al. [20] explored the evolution of artificial intelligence as an example based on the core terms in the patent literature to construct a technical network, show the distribution of technologies, understand the methods of technology evolution, and predict the future direction of technological development.

China's artificial intelligence research started relatively late but has also achieved significant results. Roberts et al. [21] introduced the history of artificial intelligence development in China over the past 40 years and presented decision-making suggestions for the further development of artificial intelligence. From the perspective of scientific methodology, Xiang X et al. [22] discovered the common mechanism of artificial intelligence development and refined the ecological information methodology. Gao F et al. [23] used the bibliometric method to conduct a co-occurrence analysis and visualization of the literature on artificial intelligence in China over the last decade. This study summarized artificial intelligence's hotspots and predicted future development trends. Based on patent data, Lundvall et al. [24] comparatively analyzed the development of the artificial intelligence industry in China, the United States, Japan, and other countries, and proposed development proposals. Based on the CNKI journal database, the analysis by Huang et al. [25] is a study in China's growing body of research on artificial intelligence education. Kai et al. [26] used the China Knowledge Network, Weipu, and Wanfang databases as source databases for their research. They used the system analysis method to discuss the development trend of artificial intelligence in the field of education in China, which catalyzed research related to the combination of artificial intelligence and education.

When organizing the relevant literature in multiple aspects, we found that current research is relatively rich, but there are still limitations in some aspects: (1) There is a lack systematic research on artificial intelligence from a macro perspective on the study of technological evolution. (2) Regarding the research on technological evolution, there are few studies that analyze the technological evolution process from the perspective of technical topics based on patent data, and there is a lack of research on deep mining of patent internal text semantics. Therefore, from the perspective of patent measurement,

this paper introduces the LDA (latent Dirichlet allocation) theme model to explore the technological evolution of artificial intelligence, intending to provide a reference for the development of the artificial intelligence industry.

## 3. Research Methodology and Data Processing

### 3.1. Workshop Air Conditioning System Model

To develop artificial intelligence technology, this paper analyzes the patent text of artificial intelligence technology using the LDA topic model from the perspective of patent technology based on Derwent patent data.

### 3.1.1. Strength Evolution of Technical Topics

Regarding the direction of text mining, the topic strength is the degree to which the theme contributes to the current time slice text. The intensity is an expression of the popularity of the relevant patent research. The greater the intensity of the topic, the more intensely researched the type of patented technology is in that time window, and the more patent applications are filed. Therefore, analyzing the change in subject matter intensity on the time axis can allow for a better observation of the research development of the relevant patent technology, which is of great significance for analyzing the evolution of patent technology [27].

This paper uses the method proposed by Li X D et al. to study the evolution of the theme of scientific journals [28], as shown in Formula (1):

$$Q(Z_{t,k}) = \frac{\sum_{d=1}^{D_t} \theta_{d,k}}{D_t} \tag{1}$$

where $Q(Z_{t,k})$ represents the intensity of the topic $t$ in the current time slice $k$, $\theta_{d,k}$ represents the probability of the $d$ topic in the $k$ document, and $D_t$ represents the number of documents on the time slice $t$.

### 3.1.2. Evolution of Technical Subject Content

(1) Calculation of Topic Similarity

The feature words in the subject of artificial intelligence technology will shift with time, and the meaning expressed by the topics will change accordingly. To analyze how the theme evolves on the timeline, it is necessary to calculate the similarity between the topics in the past two years. This paper uses the WE-cos method to calculate the similarity of the subject text, as shown in Formula (2):

$$sim(D_i, D_j) = \frac{\sum_{g \in D_i} \sum_{k \in D_j} H_{g,k} \times T(w_g) \times T(w_k)}{\sqrt{\sum_{g \in D_i} \sum_{l \in D_i} H_{g,l} \times T(w_g) \times T(w_l)} \cdot \sqrt{\sum_{k \in D_j} \sum_{m \in D_j} H_{k,m} \times T(w_k) \times T(w_m)}} \tag{2}$$

where $sim(D_i, D_j)$ represents the similarity between text $D_i$ and text $D_j$, $w$ is the word vector in the text, $H$ represents the similarity between the word vectors, and $T$ represents the TF-IDF value of the word vector. The value of *sim* is at [0, 1]; the closer to 1, the higher the similarity.

(2) Identification of topic associations

Because the topics that the LDA theme model mines each year are independent of each other, the relationship evolution is determined according to the similarity of topics in adjacent years. The rules are as follows:

$T_i^t$ is the subject in the time window $t$, and the topics in the time window $t + 1$ are sorted by the similarity with $T_i^t$, and the theme $T_j^{t+1}$ with the largest similarity is the backward theme of the theme $T_i^t$. Similarly, the topics in the time window $t$ are sorted by

similarity with $T_j^{t+1}$, and the topic $T_i^t$ with the greatest similarity is the forward theme of $T_j^{t+1}$.

In order to more clearly analyze the degree of association with the topic, the threshold is set to $\omega = 0.7$. If the similarity of a topic to the theme of its neighboring year is less than $\omega$, then there is no evolution relationship between the two themes.

(3) Determination of Theme Evolution Relationship

The theme evolution includes five situations: new life, extinction, inheritance, splitting, and merging, as shown in Figure 1. The boxes represent themes, the solid arrows represent the evolutionary relationship, and the dashed arrows represent the new or dead.

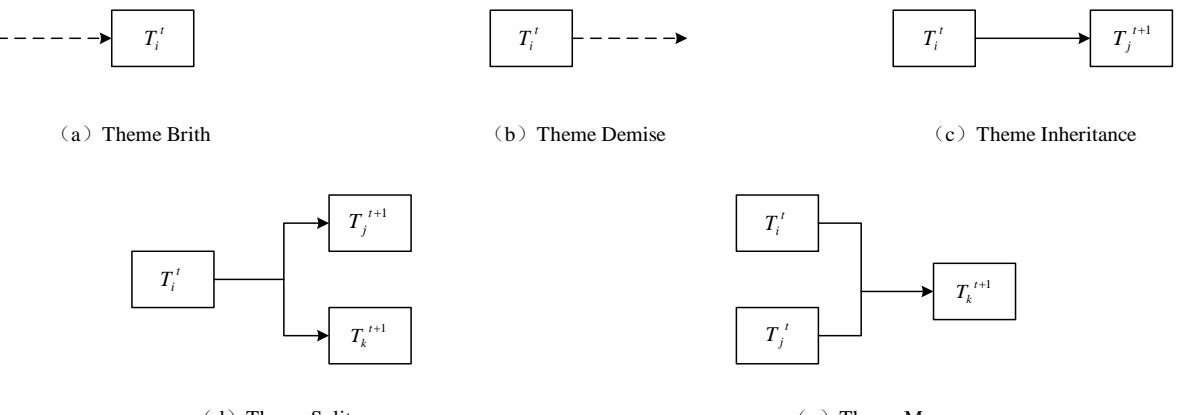

（a）Theme Brith　　　　（b）Theme Demise　　　　（c）Theme Inheritance

（d）Theme Split　　　　（e）Theme Merger

**Figure 1.** Schematic diagram of the theme evolutionary relationship.

*3.2. Workshop Air Conditioning System Model*

3.2.1. Data Collection

The patent data in this article were collected from the Derwent Innovation Index. Based on the patent classification system of the Derwent Database and previous research, this paper determines the following: TI = ("artificial intelligence *" OR "Machine learning *" OR "Pattern Recognition *" OR "Human–machine Interaction" *" OR "Depth learning *" OR "Speech Recognition *" OR "Video recognition" OR "Image Recognition *" OR "Text Recognition *" OR "smart robot *"). Patent data were collected for artificial intelligence technology, search strategy, and time. The span was set to 2011–2020, the search date was 25 May 2022, and 36,666 patent documents were obtained. The number of patents in each year is shown in Table 1.

**Table 1.** Number of patents for artificial intelligence technology between 2010 and 2020.

| Years | Number of Patents |
|---|---|
| 2011 | 562 |
| 2012 | 715 |
| 2013 | 904 |
| 2014 | 1271 |
| 2015 | 1464 |
| 2016 | 2335 |
| 2017 | 3004 |
| 2018 | 5102 |
| 2019 | 8473 |
| 2020 | 12,836 |

3.2.2. Data Preprocessing

Before the technical evolution analysis, a text cleaning of the patent data was carried out using the Python programming tool. First, the technical descriptions of the title, abstract, and keywords were extracted. Next, the patent vocabulary, academic vocabulary, stop words, special symbols, punctuation, and numbers were removed. Finally, the NTLK tool

was used in the dictionary method and combined with the WordNet Lemmatizer tool in the Python toolkit, which was used for word form reduction.

The cleaned data are displayed in .txt format on a time slice by year, and each row represents patent data. The patent data before and after cleaning are shown in Table 2. For the convenience of representation, the "Total Text Set" represents the patent data from 2011 to 2020, and the "Time Sheet Text Set" represents the patent data for each year.

**Table 2.** Data of each time slice text set before and after cleaning.

| Years | Number of Documents | Number of Words before Data Cleaning | Number of Words after Data Cleaning |
| --- | --- | --- | --- |
| 2011 | 562 | 399,935 | 74,223 |
| 2012 | 715 | 394,564 | 94,030 |
| 2013 | 904 | 467,460 | 122,431 |
| 2014 | 1271 | 717,601 | 176,621 |
| 2015 | 1464 | 707,403 | 190,808 |
| 2016 | 2335 | 1,022,635 | 306,261 |
| 2017 | 3004 | 1,213,419 | 396,898 |
| 2018 | 5102 | 1,808,530 | 719,666 |
| 2019 | 8473 | 2,389,493 | 838,059 |
| 2020 | 12,836 | 3,048,579 | 940,370 |

## 4. Results and Discussion

### 4.1. LDA Theme Mining

#### 4.1.1. Super Parameter Setting

At present, the optimal hyper parameters and settings are not clearly defined. Therefore, considering the operability of the topic, the degree of response, and the inductiveness of the results, the parameter setting is as follows: $\alpha = 0.5$, $\beta = 0.1$.

#### 4.1.2. Determination of the Number of Topics

The topic parameter K has a great influence on the determination of the final topic in the LDA model; therefore, it is necessary to determine the topic parameter K in a scientific and reasonable manner. In this paper, the more commonly used evaluation function, Perplexity, was used to determine the number of topics, and the optimal number of topics for each year and total text set was obtained, as shown in Table 3.

**Table 3.** Optimal number of topics for each time slice text set.

| Years | Optimal Number of Topics |
| --- | --- |
| 2011 | 11 |
| 2012 | 12 |
| 2013 | 12 |
| 2014 | 12 |
| 2015 | 14 |
| 2016 | 14 |
| 2017 | 15 |
| 2018 | 13 |
| 2019 | 13 |
| 2020 | 14 |
| Total text set | 26 |

### 4.2. Parameter Settings

The hyper parameter was set to $\alpha = 0.5$, $\beta = 0.1$, the number of iterations was set to 500, the iteration result was saved at 10, and the theme mining through LDA was implemented in the Python program. The subject terms were named after the comprehensive meaning of each topic feature word.

Due to space limitations, only the keywords from 2011 and the total text set are listed in Table 4.

**Table 4.** Artificial intelligence patent topic mining results.

| Years | Topic Mining Results |
| --- | --- |
| 2011 | Topic 0: Sensor, Topic 1: Process Execution, Topic 2: Alarm System, Topic 3: Speech Recognition, Topic 4: Cloud, Topic 5: User Services, Topic 6: Control System, Topic 7: Circuit and Signal, Topic 8: Image Recognition, Topic 9: Text Recognition, Topic 10: Data Analysis |
| 2012 | Topic 0: Process Execution, Topic 1: Vehicle Detection, Topic 2: Sensors, Topic 3: Data Analysis, Topic 4: Text Recognition, Topic 5: Control Systems, Topic 6: Alarm Systems, Topic 7: Speech Recognition, Topic 8: Cloud, Topic 9: Image Recognition Topic 10: Circuits and Signals, Topic 11: User Services |
| 2013 | Topic 0: Text Recognition, Topic 1: Control Systems, Topic 2: Speech Recognition, Topic 3: User Services, Topic 4: Sensors, Topic 5: Process Execution, Topic 5: Alarm Systems, Topic 7: Image Recognition, Topic 8: Data Processing, Topic 9: Circuits and Signals Topic 10: Cloud, Topic 11: Vehicle Detection |
| 2014 | Topic 0: Speech Recognition, Topic 1: Text Recognition, Topic 2: Motion Detection Devices, Topic 3: User Services, Topic 4: Image Recognition, Topic 5: Device Components, Topic 6: Machine Learning, Topic 7: Digital Signals, Topic 8: Vehicle Detection, Topic 9: Alarm Systems, Topic 10: Control Systems, Topic 11: Cloud |
| 2015 | Topic 0: Natural Language Processing, Topic 1: Neural Networks, Topic 2: Computer Vision, Topic 3: Machine Learning, Topic 4: Text Recognition, Topic 5: Motion Detection Devices, Topic 6: Cloud Intelligence, Topic 7: Video Recognition, Topic 8: Intelligent Transportation, Topic 9. Intelligent Computing, Topic 10: Digital Signals, Topic 11: Alarm Systems, Topic 12: Device Components, Topic 13: User Services |
| 2016 | Topic 0: Video Recognition, Topic 1: Cloud Intelligence, Topic 2: Motion Detection Devices, Topic 3: Smart Security, Topic 4: Smart Robotics, Topic 5: Machine Learning, Topic 6: Smart Transportation, Topic 7: Neural Networks, Topic 8: User Services, Topic 9: Device Components, Topic 10: Intelligent Computing, Topic 11: Text Recognition, Topic 12: Computer Vision, Topic 13: Natural Language Processing |
| 2017 | Topic 0: Text Recognition, Topic 1: Natural Language Processing, Topic 2: Neural Networks, Topic 3: Intelligent Healthcare, Topic 4: Intelligent Computing, Topic 5: Video Recognition, Topic 6: Computer Vision, Topic 7: Intelligent Transportation, Topic 8: Machine Learning, Topic 9: User Services, Topic 10: Smart Robotics, Topic 11: Smart Security, Topic 12: Cloud Intelligence, Topic 13: Mobile Terminal, Topic 14: Smart Finance |
| 2018 | Topic 0: Wearable Devices, Topic 1: Smart Finance, Topic 2: Smart Robotics, Topic 3: Smart Driving, Topic 4: Smart Education, Topic 5: Smart Security, Topic 6: Deep Learning, Topic 7: Smart Healthcare, Topic 8: Mobile Terminals, Topic 9: Human–Computer Interaction, Topic 10: Smart Search, Topic 11: Cloud Intelligence, Topic 12: User Services |
| 2019 | Topic 0: Smart Work, Topic 1: Smart Healthcare, Topic 2: Smart Finance, Topic 3: Wearable Devices, Topic 4: Deep Learning, Topic 5: Smart Driving, Topic 6: Mobile Terminals, Topic 7: Smart Search, Topic 8: User Services, Topic 9: Smart Education Topic 10: Smart Security, Topic 11: Human–Computer Interaction, Topic 12: Smart Robotics |
| 2020 | Topic 0: AI chips, Topic 1: Machine Learning, Topic 2: Smart Driving, Topic 3: Deep Learning, Topic 4: Smart Healthcare, Topic 5: Industrial Internet, Topic 6: Big Data Prediction, Topic 7: Cloud Platforms, Topic 8: Video Recognition, Topic 9: Speech Recognition Topic 10: Mobile terminals, Topic 11: User services, Topic 12: Human–Computer interaction, Topic 13: Image recognition |
| Total text set | Topic 0: Circuit and Signal, Topic 1: Smart Medical, Topic 2: Sensor, Topic 3: Intelligent Computing, Topic 4: Cloud Intelligence, Topic 5: Computer Vision, Topic 6: Neural Network, Topic 7: Intelligent Robot, Topic 8: Process Execution, Topic 9: Digital Signal Processing, Topic 10: User Services, Topic 11: Natural Language Processing, Topic 12: Data Analysis, Topic 13: Motion Detection Equipment, Topic 14: Deep Learning, Topic 15: Smart Driving, Topic 16: Image Recognition, Topic 17: Speech Recognition, Topic 18: Text Recognition, Topic 19: Mobile Terminal, Topic 20: Smart Security, Topic 21: Video Recognition, Topic 22: Control System, Topic 23: Human–Computer Interaction, Topic 24: Equipment Components, Topic 25: Smart Education |

*4.3. Evolution Analysis of Artificial Intelligence Technology*

4.3.1. Artificial Intelligence Technology Strength Evolution

Based on the mining results of the total text set of the artificial intelligence patent subject and the number of text sets each year, the researcher calculated the intensity value of each theme in each year according to Formula (1) and plotted the intensity change curve on the time axis. The changing trend is divided into uptrend, downtrend, and volatility trends, as shown in Figures 2–4, respectively.

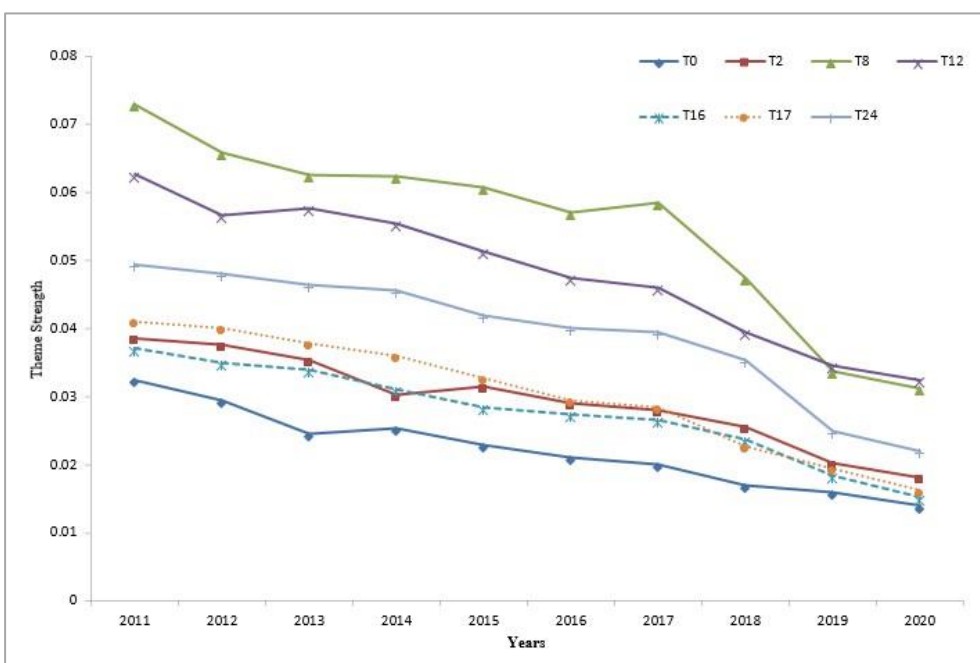

**Figure 2.** Thematic strength evolution table of the downward trend.

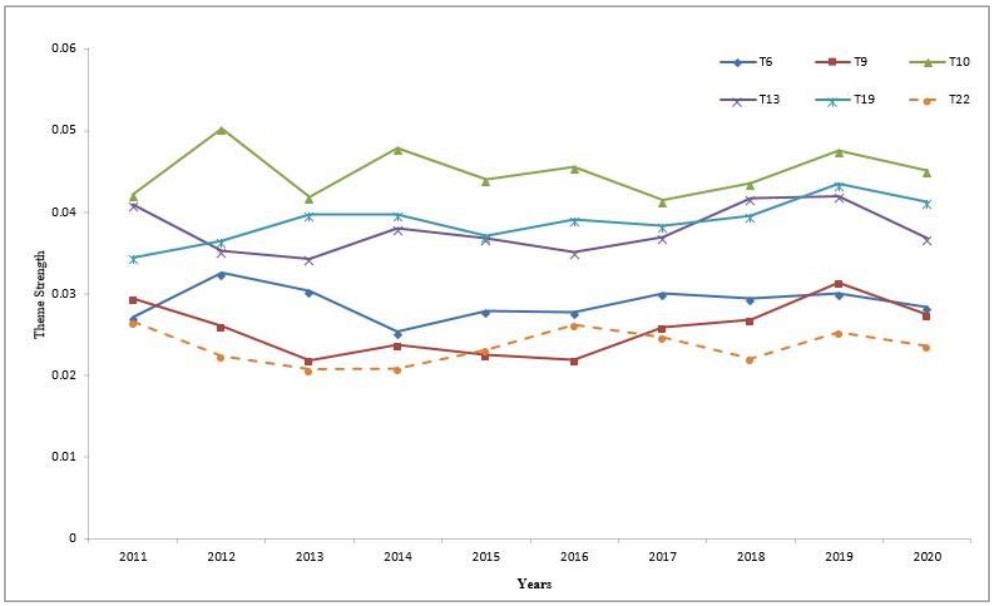

**Figure 3.** Thematic strength evolution table of the upward trend.

Figure 2 is a graph showing the change in intensity of the seven themes in the total text set, all of which have a downward trend. Figure 3 shows the intensity changes of 13 subjects in the total text set, all of which are increasing. Figure 4 is a graph showing the intensity of the subject in six trends in the total text set. Overall, basic artificial intelligence technologies, such as circuits and signals, equipment components, process execution, and data analysis have developed well in the initial stage. However, with the continuous improvement and maturity of technology, the intensity of recent research has gradually declined. Along with the vast data generated by the internet and deep learning, large-scale innovations in the fields of intelligent technology and industrial development—including intelligent algorithms, systems, and smart chips, etc.—have promoted artificial intelligence and the rapid development of speech recognition, picture video recognition, and text recognition at the technology level. Application layer evolution, smart security, smart medical, smart

driving, and other top-level artificial intelligence application technologies are on the rise. In contrast, user-related services, human–computer interactions, data security, and other technologies have also become the evolutionary direction.

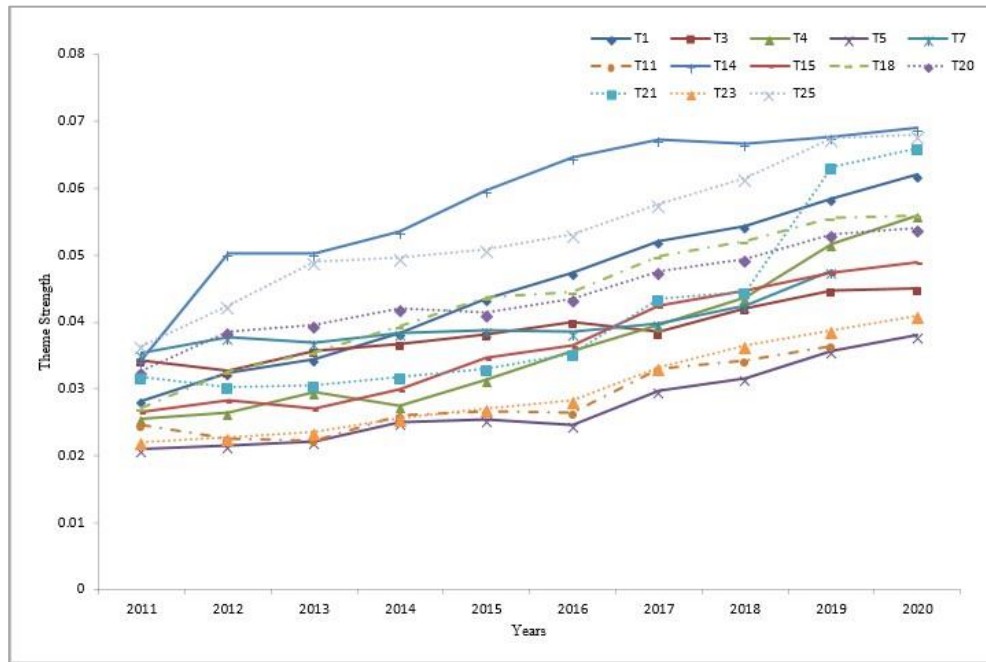

**Figure 4.** Thematic strength evolution table of the fluctuating trend.

4.3.2. Artificial Intelligence Technology Content Evolution

By combining the keywords mined from the time slice text sets from 2011 to 2020, the similarity between different topics in adjacent years was calculated according to Formula (2), and the topic similarity matrix was obtained. Under space limitations, Table 5 shows the theme similarity matrix from between 2011 and 2012 as an example. The row and column labels represent the topics in 2011 and 2012, respectively, and the values in the matrix represent the similarities of the topics.

**Table 5.** Theme similarity matrix (2011 and 2012).

|      | T0   | T1   | T2   | T3   | T4   | T5   | T6   | T7   | T8   | T9   | T10  |
|------|------|------|------|------|------|------|------|------|------|------|------|
| T0   | 0.31 | 0.88 | 0.29 | 0.54 | 0.54 | 0.53 | 0.32 | 0.37 | 0.53 | 0.49 | 0.15 |
| T1   | 0.22 | 0.68 | 0.35 | 0.33 | 0.22 | 0.44 | 0.69 | 0.83 | 0.77 | 0.48 | 0.19 |
| T2   | 0.86 | 0.52 | 0.64 | 0.32 | 0.47 | 0.42 | 0.31 | 0.72 | 0.69 | 0.77 | 0.60 |
| T3   | 0.39 | 0.45 | 0.79 | 0.85 | 0.63 | 0.33 | 0.41 | 0.68 | 0.33 | 0.43 | 0.87 |
| T4   | 0.29 | 0.32 | 0.67 | 0.16 | 0.73 | 0.61 | 0.67 | 0.25 | 0.34 | 0.85 | 0.24 |
| T5   | 0.36 | 0.38 | 0.32 | 0.72 | 0.22 | 0.73 | 0.82 | 0.72 | 0.65 | 0.55 | 0.19 |
| T6   | 0.68 | 0.17 | 0.88 | 0.31 | 0.82 | 0.24 | 0.4  | 0.45 | 0.61 | 0.33 | 0.73 |
| T7   | 0.24 | 0.54 | 0.74 | 0.88 | 0.31 | 0.84 | 0.29 | 0.21 | 0.67 | 0.80 | 0.75 |
| T8   | 0.53 | 0.53 | 0.26 | 0.40 | 0.91 | 0.63 | 0.61 | 0.26 | 0.29 | 0.39 | 0.64 |
| T9   | 0.79 | 0.31 | 0.51 | 0.61 | 0.67 | 0.83 | 0.33 | 0.64 | 0.80 | 0.20 | 0.63 |
| T10  | 0.82 | 0.68 | 0.39 | 0.82 | 0.76 | 0.26 | 0.44 | 0.87 | 0.59 | 0.75 | 0.35 |
| T11  | 0.31 | 0.68 | 0.29 | 0.54 | 0.54 | 0.93 | 0.32 | 0.37 | 0.53 | 0.49 | 0.15 |

According to the topic-related evolution relationship mentioned above, the theme similarity matrix between each year draws on the time axis shown in Figure 1, using Visio software shown in Figure 5.

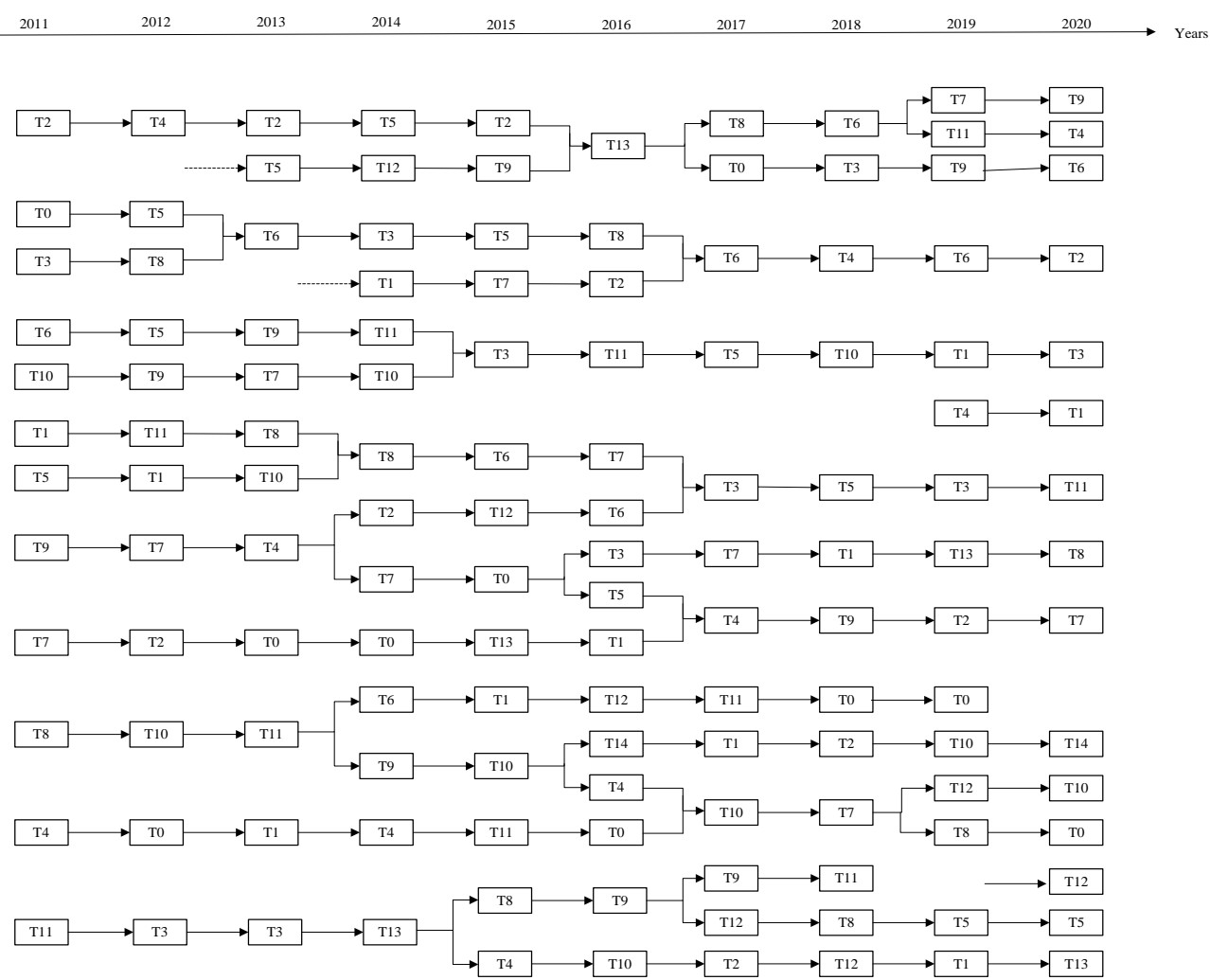

**Figure 5.** Schematic diagram of the theme evolution relationship.

According to the theme content evolution diagram (Figure 5) and the keywords of each year, the evolution of artificial intelligence technology was analyzed. The sensor technology evolved toward the motion detection device, and the device component merged first into a mobile terminal technology, and then into a wearable device technology. Data analysis technology and process execution technology merged into a machine learning technology, and then merged with neural network technology to evolve into a deep learning technology. The alarm system and circuit and signals merged and evolved into intelligent security technology. The control system and vehicle detection merged into intelligent traffic, and then merged with computer vision to evolve into intelligent driving. Image recognition and speech recognition evolved into computer vision, video recognition, and natural language processing, and merged into intelligent medical and intelligent education. Moreover, cloud technology gradually differentiated into cloud intelligence and intelligent computing, and text recognition merged into intelligent finance and intelligent search. Finally, user services were gradually categorized into human–computer interaction and intelligent robots.

Overall, the evolution of artificial intelligence technology content has gradually transitioned from the basic layer of sensors, smart chips, and algorithm models through speech recognition, image and video recognition, text recognition, deep learning, and other technologies to intelligent robots, intelligent medical care, intelligent security, intelligent education, intelligent driving, and other application areas.

## 5. Conclusions and Recommendations

### *5.1. Conclusions*

First, based on the evolution of artificial intelligence technology in the Derwent Innovation Index, this paper concludes that artificial intelligence technology is currently being upgraded from chip, sensing, and algorithm models to innovative platforms and intelligent applications. Second, big data combined with intelligent computing, deep learning, speech recognition, image and video recognition, text recognition, and other technological advancements are driving artificial intelligence technology development. Thirdly, intelligent driving, smart education, mobile terminals, intelligent robots, and other cutting-edge technologies are current research hotspots. The iterative rise of big data will promote the development of artificial intelligence in the application layer. Artificial intelligence is developing outside of the laboratory in applications such as voice translation and face recognition payments, etc. With the introduction of graphics processing unit processors, smart chips will produce subversive innovation. Likewise, intelligent robots will become more universal and generalized and can be applied to different environments. Moreover, intelligent driving will also assist in the development of automated driving.

In this context, combined with the analysis and prediction of technological evolution, as described in the previous section, countermeasures and suggestions to promote the development of AI and growth of the digital economy are proposed in terms of industrial integration, ecological systems, and independent innovation.

### *5.2. Recommendations*

#### 5.2.1. Promoting the Integration of Artificial Intelligence and Industry to Digitize Industry

Based on the spillover drive effect of AI, the interaction and intermingling of AI and other digital technologies are implemented to build intelligent new information infrastructures such as city-level operating systems and industrial internet systems to drive innovation in other technological fields. We must promote the transformation of physical industrial models, upgrade industrial structures, encourage the digital and intelligent transformation of traditional enterprises, and guide science and technology innovation enterprises to participate in the digitalization and intelligence of traditional enterprises in advance. At the same time, a government-led and market-operated model should be established. The government can integrate social capital, give full play to the multiplier effect of its funds, establish special funds for innovative service systems that promote artificial intelligence and industrial integration, and create a run of physical industry and technology integration.

#### 5.2.2. Building an AI + IoT Ecosystem and Focusing on the Industrialization of Digital Technologies

To enhance the core industrial capacity of the digital economy, it is necessary to: closely follow the Internet of Things industry; break through the limitations of artificial intelligence algorithms, arithmetic power, and arithmetic materials; promote the deep integration of the internet and artificial intelligence; build an AI + IoT ecosystem; promote the application of digital technology in the industry.

Accelerating the construction of intelligent industries, optimizing network co-design, holograms, and utilizing other intelligent technologies are important steps in extensively promoting the digitalization of industries. In the field of education, voice recognition, image video recognition, and deep learning technologies are used to establish online learning platforms. In the medical field, voice and image recognition technologies are added to realize intelligent medical treatment and carry out an automated inspection and health monitoring. In the field of transportation, with sensors and computer vision, it can analyze the surrounding environment, simulate and calculate possible situations through deep learning, and make timely judgments, thus achieving intelligent driving. In the security field, by applying the integration of video recognition, cloud computing, and other

technologies, the security system was upgraded to a defense system with active warnings and judgments.

### 5.2.3. Strengthen Core Innovation Capabilities

The key bottleneck in the development of AI in China is the lack of advanced algorithms, arithmetic power and material, core competitiveness, and key technical talents to maintain a long-term development advantage. This is a fundamentally independent innovation that strives to develop from a "follow-up" to "leading" development.

In order to organize and implement major scientific and technological projects for artificial intelligence, an enterprise's scale, management, talents and market advantages must be considered, and intelligent manufacturing should be demonstrated. As the main communities of experts, universities and enterprises should strengthen their cooperation and promote masters and doctoral degrees, and comprehensive research centers for artificial intelligence should be established.

**Author Contributions:** Data curation, methodology, software, visualization, and writing original draft, C.S. and J.W.; conceptualization, funding acquisition and project administration, and supervision, J.W.; investigation, J.W. and Y.Z.; resources, C.S. and Y.Z.; writing—review and editing, C.S. All authors have read and agreed to the published version of the manuscript.

**Funding:** This research was funded by the Major Public Welfare Science and Technology Project of Henan Province (201300311500).

**Institutional Review Board Statement:** Not applicable.

**Informed Consent Statement:** Not applicable.

**Data Availability Statement:** Not applicable.

**Conflicts of Interest:** The authors declare no conflict of interest.

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
