# Peer review of "The Evolution of Artificial Intelligence in the Digital Economy: An Application of the Potential Dirichlet Allocation Model"

_sustainability, doi:10.3390/su15021360_

Round 1
Reviewer 1 Report
This manuscript used the Derwent Innovation Index as the data source and analyzed all the patent texts through the LDA theme model. Results revealed that AI technology is upgraded from chips, sensing, and algorithms to innovative platforms and intelligent applications. However, some issues should be addressed before it can be recommended for publication.
1. In the introduction part, the authors briefly introduced the development of AI technology. More detailed background about AI should be added in this part by citing some typical references on the application of AI in various fields. Such as:
AI in agriculture (Jha, K., Doshi, A., Patel, P., & Shah, M. (2019). A comprehensive review on automation in agriculture using artificial intelligence. Artificial Intelligence in Agriculture, 2, 1-12; Dharmaraj, V., & Vijayanand, C. (2018). Artificial intelligence (AI) in agriculture. International Journal of Current Microbiology and Applied Sciences, 7(12), 2122-2128.),
AI in geotechnical engineering (s Li, K. Q., Liu, Y., & Kang, Q. (2022). Estimating the thermal conductivity of soils using six machine learning algorithms. International Communications in Heat and Mass Transfer, 136, 106139; Li, K. Q., Kang, Q., Nie, J. Y., & Huang, X. W. (2022). Artificial neural network for predicting the thermal conductivity of soils based on a systematic database. Geothermics, 103, 102416),
AI in computer science (Gupta, P. K., Saxena, A., Dattaprakash, B., Sheriff, R. S., Chaudhari, S. H., Ullanat, V., & Chayapathy, V. (2021). Applications of Artificial Intelligence in Environmental Science. Artificial Intelligence (AI), 225-240.) and so on.
2. In section 4.1, how did the authors determine the optimal number of topics?
3. In table 5, a higher level of theme similarity should be highlighted for better understanding.
Author Response
Response to Reviewer 1 Comments
Point 1: In the introduction part, the authors briefly introduced the development of AI technology. More detailed background about AI should be added in this part by citing some typical references on the application of AI in various fields. Such as:
AI in agriculture (Jha, K., Doshi, A., Patel, P., & Shah, M. (2019). A comprehensive review on automation in agriculture using artificial intelligence. Artificial Intelligence in Agriculture, 2, 1-12; Dharmaraj, V., & Vijayanand, C. (2018). Artificial intelligence (AI) in agriculture. International Journal of Current Microbiology and Applied Sciences, 7(12), 2122-2128.),
AI in geotechnical engineering (s Li, K. Q., Liu, Y., & Kang, Q. (2022). Estimating the thermal conductivity of soils using six machine learning algorithms. International Communications in Heat and Mass Transfer, 136, 106139; Li, K. Q., Kang, Q., Nie, J. Y., & Huang, X. W. (2022). Artificial neural network for predicting the thermal conductivity of soils based on a systematic database. Geothermics, 103, 102416),
AI in computer science (Gupta, P. K., Saxena, A., Dattaprakash, B., Sheriff, R. S., Chaudhari, S. H., Ullanat, V., & Chayapathy, V. (2021). Applications of Artificial Intelligence in Environmental Science. Artificial Intelligence (AI), 225-240.) and so on.
Response 1: Added typical references of artificial intelligence applications in various fields in the introduction.
Point 2: In section 4.1, how did the authors determine the optimal number of topics?
Response 2: Mentioned in 4.1.2 “In this paper, the more commonly used evaluation function Perplexity is used to determine the number of topics, and the optimal number of topics for each year and total text set is obtained”
Point 3: In table 5, a higher level of theme similarity should be highlighted for better understanding.
Response 3: Values with a similarity greater than 0.8 are displayed in bold.

Reviewer 2 Report
Journal: sustainability (ISSN 2071-1050)
Manuscript ID: sustainability-2100096
Title: The Evolution of Artificial Intelligence in The Digital Economy: An Application of the Potential Dirichlet Allocation Model.
Authors: Wang Jun , Shan Chunyi * , Zhu Yongming.
This paper aims to explore the evolutionary process of AI in the era of the digital economy to indicate the next stage of AI development. And this paper adopts the Potential Dirichlet Allocation Model to analyze the evolutionary process, and at the same time, combines the current situation of Chinese research to accelerate the development of China's AI and improve its international competitiveness.
This paper is characterized by the fact that the researchers used the LDA model (Latent Dirichlet Allocation) to explore the technological development of artificial intelligence, and this is in contrast to previous research that adopted the bibliometric method as mentioned by the authors.
In fact, as a reader, I enjoyed tracking the development of artificial intelligence, and in the recommendations made by them, which I felt added to knowledge and important for decision makers, graduate students, and researchers. These are strengths of the research, in addition to the researchers' persuasive style.
In spite of all that I have a few very minor notes:
- The abstract is very well written and expresses the paper's content and method perfectly. But I would have preferred that the objective of the paper be written in the abstract.
- Introduction and research background: I suggest that researchers mention the concept of artificial intelligence; So that if a non-professional person reads the paper, he can understand what artificial intelligence is.
- In the first paragraph of the research background; I suggest that researchers write what it refers to "[6]" instead of “is also like this [6] “, as well as mentioning the reference number.
- The first point of the research methodology; Please check the sentence highlighted in red “Aiming at the evolution of artificial intelligence technology, this paper introduces the LDA topic model into the topic mining of artificial intelligence technology in the subject of Derwent patent data. This paper intends to study artificial intelligence from two aspects: technical topic strength and technical topic content”.
- Notes on general appearance:
1. on the fourth page; Please move the point 3.2.2 at the end of the paper to the next page.
2. on page ten; Please move the point “5. Conclusions” at the end of the paper to the next page.
3. Please explain the abbreviation that appeared in the conclusion “GPU”. Is it an abbreviation for " Graphics processing unit”?
Best Wishes

Author Response
Response to Reviewer 2 Comments
Point 1: The abstract is very well written and expresses the paper's content and method perfectly. But I would have preferred that the objective of the paper be written in the abstract.
Response 1: Added relevant content in abstract: “In order to promote technological innovation and technology layout in the field of artificial intelligence, this paper analyzes the patent text of artificial intelligence technology through the LDA topic model from the perspective of patent technology subject, based on Derwent patent data.”
Point 2: Introduction and research background: I suggest that researchers mention the concept of artificial intelligence; So that if a non-professional person reads the paper, he can understand what artificial intelligence is.
Response 2: Added a description of the concept of artificial intelligence in the Introduction.
Point 3: In the first paragraph of the research background; I suggest that researchers write what it refers to "[6]" instead of “is also like this [6] “, as well as mentioning the reference number.
Response 3: Change this sentence: “The theme and documents are composed of vocabulary with different probability distributions[6].”
Point 4: The first point of the research methodology; Please check the sentence highlighted in red “Aiming at the evolution of artificial intelligence technology, this paper introduces the LDA topic model into the topic mining of artificial intelligence technology in the subject of Derwent patent data. This paper intends to study artificial intelligence from two aspects: technical topic strength and technical topic content”.
Response 4: change this sentence: “Aiming at the evolution of artificial intelligence technology, this paper analyzes the patent text of artificial intelligence technology through the LDA topic model from the perspective of patent technology subject, based on Derwent patent data.” At the same time, submit the article to the English editorial department of MDPI for text polishing.
Point 5: Notes on general appearance:
1.on the fourth page; Please move the point 3.2.2 at the end of the paper to the next page.
2.on page ten; Please move the point “5. Conclusions” at the end of the paper to the next page.
3.Please explain the abbreviation that appeared in the conclusion “GPU”. Is it an abbreviation for " Graphics processing unit”?
Response 5: The appearance format has been changed again, and the abbreviation "GPU" appearing in the conclusion has been replaced with the full spelling "Graphics processing unit".

Reviewer 3 Report
The authors used the Derwent Innovation Index as the data source to apply artificial intelligence techniques for the analysis of the evolutionary process, and the combination of the current situation of Chinese research to accelerate the development of China's AI and improve its international competitiveness. The paper is interesting, but bit must be restructured, and the language must be more clear. It was very difficult to understand the purpose of the paper at the beginning. The abstract must be more clearly presenting the background summary, the purpose, and summary of the results. The introduction must also present the main contributions of the research. The research background is fine, but the inclusion of a summary table with the advantages and disadvantages of the previous studies will benefit. Sections 3 and 4 the language must be refined. Also, section 6 is part of a discussion section, and it must be before the conclusions. Also, more recent references must be included.
Author Response
Point 1: The paper is interesting, but bit must be restructured, and the language must be more clear.
Response 1: We submit the article to the English editorial department of MDPI for text polishing.
Point 2: The abstract must be more clearly presenting the background summary, the purpose, and summary of the results.
Response 2: Added relevant content in abstract: “In order to promote technological innovation and technology layout in the field of artificial intelligence, this paper analyzes the patent text of artificial intelligence technology through the LDA topic model from the perspective of patent technology subject, based on Derwent patent data.”
Point 3: The introduction must also present the main contributions of the research.
Response 3: Added relevant content in abstract: ”Therefore, it is important to explore the evolutionary process of AI in the era of the digital economy to indicate the next stage of AI development. And this paper adopts the Potential Dirichlet Allocation Model to analyze the evolutionary process, and at the same time, combines the current situation of Chinese research to accelerate the development of China's AI and improve its international competitiveness. It is of great significance for China to carry out technological layout, discover the shortcomings of artificial intelligence technology, and promote technological innovation in artificial intelligence related fields.”
Point 4: The research background is fine, but the inclusion of a summary table with the advantages and disadvantages of the previous studies will benefit.
Response 4: A part of the literature review has been added: “By sorting out the relevant literature in multiple aspects, it is found that the current research is relatively rich, but there are still limitations in some aspects: (1) Research on artificial intelligence technology lacks systematic research on artificial intelligence from a macro level The study of technological evolution. (2) Regarding the research on technological evolution, there are few researches on analyzing the technological evolution process from the perspective of technical topics based on patent data, and there is a lack of research on deep mining of patent internal text semantics.”
Point 5: Sections 3 and 4 the language must be refined.
Response 5: The language has been refined and we submit the article to the English editorial department of MDPI for text polishing.
Point 6: Section 6 is part of a discussion section, and it must be before the conclusions.
Response 6: Combining section 5 and section 6 into one section,named “Conclusions and Recommendations”.
Point 7: More recent references must be included.
Response 7: Added relevant references from recent years.

Reviewer 4 Report
Reviewer’s Comments
A minor revision is being suggested for Manuscript No.: Sustainability-2100096, titled, "The
Evolution of Artificial Intelligence in The Digital Economy: An Application of the Potential
Dirichlet Allocation Model". The following are the observations/corrections that needs to be
incorporated:
1. Abstract, Line 13-14: “From the patent.... ”, sentence must be rewritten appropriately.
2. Introduction, Line 41-42: “And this paper…..”, sentence must be rewritten
appropriately.
3. Authors must avoid the unnecessary capitalization, suggested to check in the complete
manuscript and revise.
4. Research background: Author must add some more relevant literature and identify the
research gap, afterwards appropriate objectives must be formulated and should be
properly mentioned in the last paragraph of the same section.
5. Research methodology, Line 113-114: “Li Xiangdong and Zhang Jiao”,
appropriate style of citation must be followed.
6. Results and discussion: Currently the Results and discussion contains only the results,
discussion is totally missing. Authors are suggested to identify the similar work
published recently and compared the obtained results with the existing work.
Conclusions: Authors must rewrite the conclusions and summarize major findings in
150-200 words max. Conclusions must contain major findings of your work in numbers
and major takeaway for the reader.
7. Recommendations and Practical Implications: This section contains major two things
(i) Recommendations: Which indirectly the future work, it can be integrated with the
conclusions with a separate paragraph of 50-100 words max.
(ii) Practical Implications: There are the points majorly related to the discussion based
on the existing literature (case based analysis), it can be integrated with the results and
discussion section with appropriate citation (which is missing in current form).
I wish authors a great success.

Author Response
Point 1: Abstract, Line 13-14: “From the patent.... ”, sentence must be rewritten appropriately.
Response 1: Change this sentence: “In order to promote technological innovation and technology layout in the field of artificial intelligence, this paper analyzes the patent text of artificial intelligence technology through the LDA topic model from the perspective of patent technology subject, based on Derwent patent data.”
Point 2: Introduction, Line 41-42: “And this paper…..”, sentence must be rewritten appropriately.
Response 2: Change this sentence: “This paper analyzes the patent text of artificial intelligence technology through the LDA topic model from the perspective of patent technology subject, based on Derwent patent data. It is of great significance for China to carry out technological layout, discover the shortcomings of artificial intelligence technology, and promote technological innovation in artificial intelligence related fields.”
Point 3: Authors must avoid the unnecessary capitalization, suggested to check in the complete manuscript and revise.
Response 3: Already checked the revision in full.
Point 4: Research background: Author must add some more relevant literature and identify the research gap, afterwards appropriate objectives must be formulated and should be properly mentioned in the last paragraph of the same section.
Response 4: Revisions have been made, see research background.
Point 5: Research methodology, Line 113-114: “Li Xiangdong and Zhang Jiao”, appropriate style of citation must be followed.
Response 5: Revisions have been made:”Li X D,et al.”.
Point 6: Results and discussion: Currently the Results and discussion contains only the results, discussion is totally missing. Authors are suggested to identify the similar work published recently and compared the obtained results with the existing work. Conclusions: Authors must rewrite the conclusions and summarize major findings in 150-200 words max. Conclusions must contain major findings of your work in numbers and major takeaway for the reader.
Point 7: Recommendations and Practical Implications: This section contains major two things
(i) Recommendations: Which indirectly the future work, it can be integrated with the conclusions with a separate paragraph of 50-100 words max.
(ii) Practical Implications: There are the points majorly related to the discussion based on the existing literature (case based analysis), it can be integrated with the results and discussion section with appropriate citation (which is missing in current form).
Response 6 and 7: Combining section 5 and section 6 into one section,named “Conclusions and Recommendations”.At the same time, the language was streamlined and revised.

Round 2
Reviewer 3 Report
The manuscript was improved, and it can be accepted.